# Health outcomes, healthcare use and development in children born into or growing up in single-parent households: a systematic review study protocol

Irina Lut ,[1] Jenny Woodman ,[2] Alice Armitage ,[1] Elizabeth Ingram ,[3] Katie Harron ,[1] Pia Hardelid [1]

¹UCL GOS Institute of Child Health, London, UK
²UCL Institute of Education, London, UK
³UCL Department of Applied Health Research, London, UK

**Correspondence to**
Ms Irina Lut; i.lut@ucl.ac.uk

## ABSTRACT

**Introduction** Up to a quarter of all children globally live in single-parent households. Studies have concluded that children who grow up with continuously married parents have better health outcomes than children who grow up with single or separated parents. This is consistent for key health and development outcomes including physical health, psychological well-being and educational attainment. Possible explanations include higher poverty and time limitations of parental engagement within single-parent families, but these only represent a narrow range of mechanisms. We aim to identify and synthesise the evidence on how being born into and/or living in a single-parent household compared with living in a two-parent household as a child impacts health and development outcomes, healthcare use and factors that may be driving differences.

**Methods and analysis** We will search PubMed, Scopus and ERIC and adapt our search terms for search engines and grey literature sites to include relevant conference abstracts and grey literature. We will restrict results to English language publications from 2000 to 2020 and screen against inclusion criteria. We will categorise main outcomes into five groups of outcomes: birth outcomes, mortality, physical health, mental health and development, and healthcare use. We will use the Newcastle-Ottawa Scale to assess the methodological quality of studies. Narrative synthesis will form the primary analysis in the review. Synthesis of effect estimates without meta-analysis will follow the Synthesis Without Meta-analysis guidelines.

**Ethics and dissemination** All documents used are publicly accessible. We will submit results to a peer-reviewed journal and international social science conferences. We will communicate results with single-parent groups and relevant charitable organisations. This review will also be included in IL's PhD thesis.

**PROSPERO registration number** CRD42020197890.

### Strengths and limitations of this study

► This review will fill an evidence gap on the drivers and protective factors that influence the health and development of children growing up in single-parent households.
► A robust methodology and extensive search strategy will support clear results to inform policies and interventions to support single-parent households.
► Findings from included studies will likely be heterogeneous in terms of definitions of single-parent households, and definitions and measurements of outcomes, which may preclude meta-analysis.

parent.[2] The proportion of single-parent families in Great Britain has remained stable over the last 20 years[3] following an increase of single-parent families between 1970 and 1995.[4] The key reasons for this increase were rising levels of divorce and partnership breakdown during the 1970s and 1980s[2 5] and an increase in the number of births to single women since the mid-1980s.[4 6 7]

Multiple studies have concluded that children who grow up with continuously married parents have better outcomes than children who grow up with single parents or children whose parents separate during childhood.[8–10] This is consistent for key health and development outcomes including physical health,[11] psychological well-being[12] and educational attainment.[13]

A systematic review of maternal marital status and birth outcomes from 2010 has summarised the current literature on risks of an infant being born with low birth weight (<2500 g), preterm birth (<37 weeks' gestational age) or small for gestational age (below the 10th percentile for babies of the same gestational age) among married and unmarried women.[14] Findings identify significantly increased odds of low birth weight, preterm

## INTRODUCTION

Between 10% and 25% of children in member countries of the Organisation for Economic Co-operation and Development (OECD) live in single-parent households.[1] In Great Britain, 15% of all families are headed by a single

birth and small for gestational age births among unmarried women compared with married women. A further systematic review found that children in single-parent households have higher body mass index and obesogenic behaviours such as insufficient physical activity and increased television viewing time, compared with children living with two parents.[15]

Socioeconomic factors such as income, occupation and education (also referred to as socioeconomic status or SES) are strongly associated with both parental and child well-being.[16] Children of single parents are at higher risk of living in poverty and deprivation compared with children growing up in coupled families.[17] In 2018, 49% of children in single-parent families lived in poverty in England compared with 25% of children in coupled families.[18] Women, who head approximately 88% of single-parent families globally,[19] have lower earnings than men on average due to gender wage gaps and salary penalties for motherhood.[20] Additionally, though the majority of single parents in OECD countries are in some form of paid employment, single parents are more likely to work in occupations with lower earning potential and job security, and must balance work responsibilities with childcare.[21] While it is clear that single-parent families are disadvantaged with respect to SES and health outcomes, it remains unclear whether SES fully explains differences in outcomes for children of single mothers compared with children with two parents. Other potential mechanisms linking single parenthood to a higher risk of adverse child health outcomes have been reported, including parenting stress, lack of social networks and support, and social stigma which can influence maternal mental health and effective parenting.[11 22] It is difficult to examine the extent to which each factor may individually affect the association between living with a single parent and adverse health in children, particularly since the likelihood of becoming a single mother is very strongly associated with SES. Women with lower SES (based on their father's occupation) were up to six times more likely to become single mothers in a study of three large British surveys.[23] The rate of relationship breakdown resulting in single motherhood was found to be almost double among women in unskilled work compared with women in professional or managerial roles.[23]

A substantial body of research also exists on the health impact of parental relationship breakdown (a mode of entry into single parenthood) or reforming on child health.[24 25] However, a review of literature up to 2005 concludes that children with two continuously married parents attain better cognitive and emotional outcomes compared with children with only one biological parent in the household,[9] and this is more plausibly explained by higher deprivation and lower education among single mothers than by the impacts of a parental relationship breakdown.[4]

Research on single parenthood has focused largely on single mothers. More often than men, women parent alone due to breakdown of a relationship or pursue parenthood without a partner from the point of conception.[4] Research on the quality and quantity of fathering exists but has tended to focus on the impact of father absence rather than single-father families.[26] There are well-described challenges to capturing fathers in research exploring the impact of parents on children's outcomes[27 28]; even less is known about different characteristics or subgroups of single fathers than single mothers.

Historically, official statistics agencies have relied heavily on marital status to define single motherhood. The definition of a family, which has been based on blood or marriage ties in countries like the USA, drives the classification of one or two-parent families.[29] Unmarried women who have had children with cohabiting partners and lived in a two-parent household have previously been grouped with single mothers, leading to inflation of the number of families that appeared to be led by non-partnered women.[4] This is despite cohabiting households with children being one of the fastest growing family forms between 1980s and 2000s in the UK and other countries.[3] Single mothers who have separated from a partner, either via divorce or relationship breakdown, are likely to be different from women with no partner who become mothers[4] and report different parenting experiences.[30] Capturing nuances in single-parent households may be critical in understanding why children of single mothers have poorer outcomes compared with children of coupled mothers and identifying protective factors. Distinguishing between different types of and all routes into single parenthood is important as family structures have become more complex and new non-traditional family forms are being recognised.[4 31] Definitions and terminology matter to make sure we understand the comparisons we are making between groups and to ensure that negative or stigmatising narratives associated with single motherhood are not perpetuated.[32]

In this systematic review we will compare a range of health and development outcomes among children living in single-parent households and children living in coupled-parent households, and identify factors that may be driving differences. We will focus particularly on children who remain in single-parent versus continuously coupled families but also include comparisons with cohabiting and married couple families, or subgroups of single-parent families (separated single mothers, never-married single mothers by choice, single fathers) where available. We aim to fill gaps in evidence by exploring whether health disparities between children of single parents and children of coupled parents persist after taking into account socioeconomic characteristics, presenting findings that explain the differences and reporting protective factors that allow children to be healthy in a single-parent family. Our findings will highlight areas where policy change or public health interventions might help improve health of the large numbers of children living in single-parent households.

## METHODS AND ANALYSIS

### Aims and research questions

The aim of this review is to systematically identify and synthesise the evidence on how being born into and/or living in a single-parent household as a child impacts health outcomes, healthcare use and development outcomes, compared with living in a two-parent household, and factors that may be driving differences.

This systematic review will answer the following questions:

1. How do health, healthcare use and development outcomes compare among children and young people (less than 18 years old) growing up in single-parent and coupled-parent households?
2. What factors influence any observed differences in child health, healthcare use and development outcomes between children of single parents versus coupled parents?

### Searches

We will search for the concepts 'single parents' AND 'child health' OR 'child development' outcomes using indexed Medical Subject Headings and free text terms, restricting results to English language publications from 2000 to 2020. We will search three databases which index medical, social science and education research: PubMed, Scopus and ERIC. We will identify additional relevant results through backward and forward citation searching and grey literature search engines. We provide the full list of search concepts and terms in online supplemental appendix 1 (carried out on 15 July 2020). We will adapt our search terms for search engines like Google Scholar and Scirus, and refer to the Canadian Agency for Drugs and Technologies in Health guidance for links to grey literature sites relevant in the UK context, to search for conference abstracts and grey literature or additional peer-reviewed articles.

Two researchers (IL and AA or IL and EI) will independently screen all results based on title and abstract and further screen full texts for inclusion. A third reviewer (PH or JW) will resolve any discrepancies.

### Inclusion and exclusion criteria

The population of interest is children who have experienced living in a single-parent household at any time during childhood (aged less than 18 years) and have at least one of the health outcomes measured in the study before the age of 18 years. We will include studies if the single parent (either mother or father) is living with dependent children and does not have a partner living in the same home. Only studies with enough information to identify the single-parent exposure group as we have defined it here will be included; studies where the exposure groups are married versus unmarried, without further specification of cohabitation status of parents, will be excluded.

Parents may transition in and out of relationships with different people.[4 33] While they may be consistently partnered, changes in family structure (also referred to as family instability) have also been shown to negatively impact child outcomes.[24] In this review, we will include studies that categorise children as 'ever having lived in a single-parent family during childhood' if the health impact of living with a single parent is also examined.

A substantial body of work shows that parental conflict and poor marital quality adversely affect behavioural outcomes, anxiety and depression, and emotional security in children and adolescents.[34 35] However, in this systematic review we will exclude studies that focus exclusively on the health effects of parental relationship breakdown or quality without investigating the effects of single parenthood.

We will include studies with any definition and measure of the five types of outcomes. Studies employing quantitative study designs such as cohort, cross-sectional and case–control studies will be included. A range of study types will provide a comprehensive view of the literature with a mix of well-powered studies, longitudinal data points and objectively measured outcomes.

### Outcomes

We present the main outcomes in this review in five groups:

1. *Birth outcomes*: including birth weight, low birth weight (<2500 g), very low birth weight (<1500 g), gestational age, small for gestational age (<10th percentile), preterm birth, congenital anomalies.
2. *Mortality outcomes*: including stillbirth, perinatal mortality, child mortality.
3. *Physical health outcomes*: including nutrition, weight, oral health, motor skills.
4. *Mental health and development outcomes*: including disruptive behaviour, substance abuse, anxiety or depressive disorders, autism spectrum disorders, psychosis, self-harm and suicidality, cognitive abilities (problem solving, memory, language/communication, early years' educational attainment), social-emotional development (personal-social skills).
5. *Healthcare use outcomes*: including any hospital admission (planned or emergency), vaccinations, visits to primary care, contact with health visitors or well-child checks.

### Data extraction and management

For each included study, we will extract information on study authors and date of publication, study setting (country and its World Bank income group classification if available) and period (year), study design (including selection criteria, number of participants and analysis, causal claims), timings and definition of single parenthood and outcomes of the study. If available, we will additionally extract information about confounding variables that were controlled for, variables reported as effect modifiers of the relationship between single parenthood and child health and development and variables that act as measures of SES (eg, use of income support or tax credits,

employment or access to health insurance). All management of included publications and extracted data will be done within the EPPI-Reviewer software.

Studies will be grouped by exposure groups (single-parent vs coupled-parent household) and main outcomes. Definitions of single parenthood may vary across studies and exposure groups are likely to differ by the type, timing and duration of parental relationship status. In most studies, the exposure is expected to be self-reported exposure or obtained from an administrative data source. Authors will be contacted if the time parameters of single parenthood are not clear from the published work.

Outcomes will be grouped into the five main outcome groups defined above. We will use the Newcastle-Ottawa Scale to assess the methodological quality of studies.[36] We will use the Risk of Bias in Non-randomised Studies of Interventions tool for assessing risk of bias in non-randomised studies.[37]

### Synthesis and meta-analysis

Given the range of outcomes and the likely diversity in the way single-parent households are defined across studies, we expect that the included studies will be too heterogeneous to carry out meta-analyses. Narrative synthesis will therefore form the primary analysis in the review. To carry out a robust narrative synthesis, we will incorporate the four main elements described in the Economic and Social Research Council Methods Programme guidance on the conduct of narrative synthesis for systematic reviews.[38] We will follow the nine reporting items of the synthesis of effect estimates without meta-analysis from the Synthesis Without Meta-analysis guidelines[39] and discuss the limitations of the synthesis methods used.

We will first report how single parents are defined in each study, creating a typology based on the literature which will inform how studies are categorised for synthesis or subgroup meta-analysis. We then aim to synthesise results by distinct types of family forms (eg, single or coupled parents, married or cohabiting, single mother or single father) and the age(s) at which a child is living in a single-parent household. Where the age of the child during the exposure period or the duration of the exposure period is clearly reported, subgroup analyses will be carried out by age at which the child lived in a single-parent household and by the length of exposure to single parenthood. Subgroup analyses will also be carried out separating single mothers and single fathers. Additional subgroup analyses or special attention in reporting will be considered to take into account SES (based on SES indicators as available) and country context (based on World Bank income group classification) that could influence the association between single parents and child health.

For each of the five main outcome groups, we will summarise the health and development outcomes most commonly reported and report any significant differences between children living with single parents and children living with coupled parents at any point during childhood. If any differences are reported between children living in different subgroups of single-parent households (separated single mothers vs never-married single mothers by choice vs single fathers), these will also be described. This will address our first research question. From included studies, we will identify factors such as employment or social support that influence differences in outcomes between children in single-parent households and children in coupled-parent households. Identifying potential mechanisms impacting the relationship between single parenthood and child health (eg, access to healthcare or family income) may inform policy change or intervention targeting improvements in child health and development outcomes. This will address our second research question.

Should at least three studies employ the same design, and have similar exposure groups and outcomes, a decision will be made by the review team on whether meta-analysis is appropriate. A heterogeneity test ($I^2$ statistic) may be used to describe the percentage variability between studies and confirm whether it is reasonable to pool studies that appear comparable. Studies that have comparable exposures or outcomes but that are categorised as low quality based on the Newcastle-Ottawa Scale will not be included. Should a meta-analysis be appropriate, we will pool data using the DerSimonian and Laird random effects models.[40] We will calculate adjusted measures of association (such as ORs, hazard rates and relative risk) presented using logarithmic scales for dichotomous categorical outcomes and standardised mean difference for continuous outcomes. We will carry out separate meta-analyses for unadjusted and adjusted effect sizes to better understand the effects of confounding variables on the association between single parenthood and child outcomes. We will visualise results as forest plots. We will use funnel plots to assess publication bias.[41]

### Patient and public involvement

No patients were involved in the development of this protocol.

## ETHICS AND DISSEMINATION

No requests for ethical approval have been made given that all documents used are publicly accessible. We will submit results to a peer-reviewed journal for publication and international social science conferences. We will communicate results with single-parent groups and relevant charitable organisations. This review will also be included in IL's PhD thesis.

**Contributors** The protocol was conceived by all the authors, written by IL in collaboration with PH, KH and JW and reviewed by AA and EI prior to submission.

**Funding** This work is supported by the NIHR Great Ormond Street Hospital Biomedical Research Centre. IL is funded by the Medical Research Council (grant reference MR/N013867/1).

**Competing interests** None declared.

**Patient consent for publication** Not required.

**Provenance and peer review** Not commissioned; externally peer reviewed.

**Open access** This is an open access article distributed in accordance with the Creative Commons Attribution 4.0 Unported (CC BY 4.0) license, which permits others to copy, redistribute, remix, transform and build upon this work for any purpose, provided the original work is properly cited, a link to the licence is given, and indication of whether changes were made. See: https://creativecommons.org/licenses/by/4.0/.

**ORCID iDs**
Irina Lut http://orcid.org/0000-0002-9075-648X
Jenny Woodman http://orcid.org/0000-0002-9403-4177
Alice Armitage http://orcid.org/0000-0001-6972-3651
Elizabeth Ingram http://orcid.org/0000-0002-0354-4551
Katie Harron http://orcid.org/0000-0002-3418-2856
Pia Hardelid http://orcid.org/0000-0002-0154-1306

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
