## [Reviewer comments · BMJ Open]

ARTICLE DETAILS

TITLE (PROVISIONAL)	Health outcomes, healthcare use and development in children born into or growing up in single-parent households: A systematic review study protocol
AUTHORS	Lut, Irina; Woodman, Jenny; Armitage, Alice; Ingram, Elizabeth; Harron, Katie; Hardelid, Pia

VERSION 1 – REVIEW

REVIEWER	Nain-Feng CHU TSGH, NDMC, Taiwan, ROC
REVIEW RETURNED	04-Sep-2020

GENERAL COMMENTS	bmjopen-2020-043361 This is an interesting protocol that evaluates health outcomes (including birth outcomes, mortality, physical health, mental health and development) and health care utilization among children who grow up in continuous married parents and single/separated parents. However, there are some points could be addressed more clearly in this protocol. 1. Continuous married parents – are quality or relationships between parents in these family considered? Since some of the parents will keep the married status for their children but have a bad relationship.2. There are five groups of outcomes in this protocol – however literature on some of the outcomes such as mental health and development are limited in some under-developed countries.3. Healthcare utilizations also vary from countries to countries based on the health insurance status in their countries.4. How this study can be adjusted for the socioeconomic characteristics to evaluate health disparities among children based on this protocol?5. Regarding the measurement of exposure (single parent household and the length of exposure to single parenthood) – how are the validity and reliability of these data?6. The main problem of this protocol may be due to publication bias and the traditions or customs in different countries.7. How can this study highlight the policy change and public health interventions in health outcomes and healthcare use?8. How could the results of these studies be summarized without meta-analyses?9. I am wondering how this study can answer the factors associated with the difference of health outcomes and healthcare use – based on the systematic review data.
---

REVIEWER	Marcia Gibsom MRC/CSO University of Glasgow Social and Public Health Sciences Unit, United Kingdom
REVIEW RETURNED	11-Nov-2020

GENERAL COMMENTS	Many thanks for the opportunity to review this systematic review protocol. Increasing understanding of the mechanisms linking lone parenthood to less favourable child outcomes is a very important research question, and I am glad to see that a review of this topic is being conducted. The protocol is well written and the research questions are clear. I have several comments below which I hope will help to improve the protocol. The authors note that health is better in children with continuously married parents than for children with a single parent or separated parents. However, it is common to transition in and out of relationships, usually with new partners. Will outcomes for children in such households be considered, or will study participants be defined in terms of having only one biological parent in the household regardless of subsequent relationship transitions? The authors note that poverty is higher among single parent families, but there is no discussion of why this is the case. Higher poverty and less time with children are mentioned as just two of a range of possible mechanisms linking lone parenthood to adverse child outcomes in the abstract. The background section focuses on the role of parental SES both prior to and after becoming single parents. However, the literature has identified many other potential mechanisms linking lone parenthood with adverse child outcomes, including lack of social support, stigma, role strain, and unemployment. Studies I am aware of tend to find that negative health outcomes are not entirely attenuated when SES is controlled for, leaving some uncertainty about the other factors contributing to poor outcomes. Some discussion of the range of potential mechanisms, and the nature of the evidence for them, would be helpful for framing the review. The authors note the challenges involved in studying lone fathers, but do not explicitly state whether studies of lone fathers will be included in the review. Given that they seem to be quite a distinct population, would it be appropriate to combine studies of lone mothers and lone fathers? Again, the authors refer to the difficulty of defining and sometimes identifying lone parents, and how different types of lone parent family may influence outcomes, but do not discuss how they will approach this in the studies. Will an attempt be made to establish how LPs are defined in each of the studies, and if so whether/how these differences do impact on child outcomes? For instance, as far as I am aware the USA has historically defined cohabiting mothers as single parents, whether they reside with the biological father or a new partner. In Sweden, joint custody is now the default position, and it is rather difficult to identify who is the 'lone parent' in such circumstances. Differences such as this would have to be considered when comparing outcomes across studies from different countries. Was an information scientist involved in developing the search strategy? I would suggest including the term 'solo parent' as that is
---

	commonly used in Australia. I'm unsure whether it might be helpful to include terms for specific outcome measures (e.g. Child Behaviour Checklist etc.). Given that the likelihood of becoming a lone parent is much higher for people with lower SES, how do the authors intend to deal with causality in the included studies? I would assume studies which control for parental SES prior to becoming parents would be required to account for selection effects, although I am not personally aware of many studies which do so. There is also evidence that people with mental health issues are more likely to become lone parents, again leading to selection effects. Related to this, there is no discussion of study design in the inclusion criteria. Will cross-sectional studies be included, or only longitudinal studies which can shed light on causality? Would qualitative studies be considered, as they can be particularly helpful for identifying mechanisms? Even if any study design will be included, it would be helpful to have some discussion about the advantages and limitations of different methods. There is no mention of geographical inclusion criteria. Will studies from any country be included or will they be restricted to high-income countries? Cultural and contextual differences with lower income countries may be quite challenging to account for in the synthesis. Do the authors intend to include outcomes such as substance abuse, self-harm or suicidality in the mental health domain? How will contextual differences be dealt with? Some studies have shown that outcomes for lone parents vary across countries with differing welfare regimes, and also that economic and labour market changes can influence outcomes for lone parents. These provide insights into possible mechanisms linking lone parenthood and other outcomes, and indeed potential intervention points for addressing adverse outcomes. I'm not clear why a risk of bias tool developed for systematic reviews is being used to inform the risk of bias assessment. Perhaps the ROBINS-I tool would be helpful for assessing non-randomised studies. Weighted mean differences will be used for continuous outcomes. Given that reported outcomes such as depression are likely to use heterogeneous scales, I would suggest that standardised mean differences would be more appropriate. These can also permit inclusion of several heterogeneous outcomes from single studies in forest plots even where meta-analysis is not possible. NB line 26 'four' should read 'five'.
--	---

VERSION 1 – AUTHOR RESPONSE

Reviewer: 1

This is an interesting protocol that evaluates health outcomes (including birth outcomes, mortality, physical health, mental health and development) and health care utilization among children who grow up in continuous married parents and single/separated parents. However, there are some points

could be addressed more clearly in this protocol.

1. Continuous married parents – are quality or relationships between parents in these family considered? Since some of the parents will keep the married status for their children but have a bad relationship.

The reviewer is correct in identifying this complexity. In the third paragraph of the inclusion/exclusion criteria (p6, lines 280-282), we acknowledge that marital quality has been shown to impact on internalizing and externalising behaviour, anxiety and depression, and feelings of security in children and teens. This shows that in the same way that not all single-parent families are the same, not all married couple families are equally functional. However, our focus is to review the literature on the association between family structure and child health, rather than examine the quality of relationships within coupled parents and its impact on child health (p6, lines 282-284).

2. There are five groups of outcomes in this protocol – however literature on some of the outcomes such as mental health and development are limited in some under-developed countries.

We have chosen a broad range of outcomes and settings to be included at the screening stage and will assess the best way to present these results once the final selection is made for inclusion. We will extract information on the country where each included study was carried out, as stated in the data extraction section of the protocol (p6, line 312) and if important gaps in evidence are noted, this will be included in the discussion section when we write up the findings of the systematic review.

3. Healthcare utilizations also vary from countries to countries based on the health insurance status in their countries.

Thank you for highlighting this; this is correct. At this stage we will include all settings and should information on health insurance/access to healthcare be available, we will extract this information as an indicator for socioeconomic status. We have now specified in the data extraction section of the protocol that we will collect this information if available (p6, lines 315-322). In the second paragraph of the synthesis section of the protocol, we have also stated that sub-group analyses will be carried out if the data is available (p7, lines 354-358).

“Sub-group analyses will also be carried out separating single mothers and single fathers. Additional sub-group analyses or special attention in reporting will be considered to take into account socioeconomic status (based on SES indicators as available) and country context (based on World Bank income group classification) that could influence the association between single parents and child health.”

4. How this study can be adjusted for the socioeconomic characteristics to evaluate health disparities among children based on this protocol?

As part of the review, we have now specified that information on any confounding variables including reported socioeconomic measures will be collected during data extraction (p6, lines 315-322). In our narrative synthesis, we will summarise measures and indicators of socioeconomic status that have been included in adjusted analyses and carry out sub-group analysis based on SES and World Bank country income classification, as described in the synthesis section (p7, lines 354-358).

Where meta-analysis is possible, we have specified that we will calculate adjusted measures of effect (ie. odds ratios, hazard rates, relative risk ratios) taking into account confounding variables as outlined in our synthesis section (p8, lines 399-401). Additionally, we will do separate meta-analyses for unadjusted and adjusted effect sizes, to better understand the confounding effect of SES (p8, lines 401-404).

5. Regarding the measurement of exposure (single parent household and the length of exposure to single parenthood) – how are the validity and reliability of these data?

Thank you for this question. Based on initial searches, we expect the exposure variable will be primarily based on administrative records (for example birth certificates, tax records etc) or self-reported by the mother. Administrative records are usually derived from legal documents and indicate eligibility for public services; therefore, we expect the accuracy to be high. We have found no study measuring the accuracy of self-reported single-parent status to indicate the validity and reliability of this type of data. We do acknowledge that certain potentially unrepresented nuances, such as the involvement of non-residential parents may influence the association between single parenthood and child outcomes.

In the data extraction section of the protocol, we have further clarified that information on the definition of single parenthood will be extracted (p6, line 314) and that most studies are expected to have self-reported parental relationship status or obtain the data from administrative sources (p7, lines 328-329).

6. The main problem of this protocol may be due to publication bias and the traditions or customs in different countries.

Thank you, we agree with the reviewer's point. To mitigate some of the potential publication bias, we have stated in the methods section that grey literature will be searched to capture any relevant literature which does not appear in peer-reviewed journals (p5, lines 242-246). Additionally, we have now added to the synthesis section that we will use funnel plots to assess publication bias of included studies (p8, lines 404-405). Funnel plots depict the effect estimates of individual studies against a measure of study size and can indicate publication bias if results from small studies do not scatter evenly along the bottom, creating an asymmetrical funnel plot.

See <https://journals.sagepub.com/doi/pdf/10.1177/1536867X0400400204>

Regarding different customs, as we state in the methods section, we will extract information about study settings and will consider sub-group analysis (for example separating low-income and high-income countries) if this data is available (p6/7, lines 318-322). We will acknowledge this when we

write up results. However in-depth discussion of the way cultural context might impact on the association between single parenthood and child health is outside the scope of this review. Any important gaps in evidence will be reported.

7. How can this study highlight the policy change and public health interventions in health outcomes and healthcare use?

Since the early 1990s, the importance of systematic reviews for evidence-based decision making within the health sector has emerged, influencing clinicians' practice, resource allocation, as well as design and implementation of interventions. (see <https://www.ncbi.nlm.nih.gov/pmc/articles/PMC3214717/>). By synthesising differences in outcomes in children of single versus coupled parents and identifying factors which may be acting as effect modifiers in the association between single parenthood and child health, we believe that this review can highlight health needs of single-parent families and areas where interventions may be appropriate to counterbalance any negative effects of living in a single-parent family that may be reported. For example, barriers to accessing primary healthcare for certain families might affect the severity for cause of admission for hospital attendance or vaccination rates. We have clarified this in the synthesis section (p7/8, lines 368-390).

8. How could the results of these studies be summarized without meta-analyses?

In the synthesis section of the protocol, we outline that we will be using narrative synthesis as the primary form of summarising results, following the guidelines for Synthesis Without Meta-analysis (SWiM) in systematic reviews (p7, lines 342-345). This involves:

- (1) providing a description and rationale for the groups used in the synthesis
- (2) describing a standardised metric (used to measure effect) for each outcome
- (3) describing and justifying the synthesis methods for each outcome where meta-analysis is not possible
- (4) reporting the criteria used to prioritise results for summary and synthesis
- (5) stating the methods to examine heterogeneity when meta-analysis is not possible
- (6) describing the methods used to assess the certainty of synthesis findings
- (7) describing the methods used to present effects
- (8) providing a description of the synthesised findings along with certainty of findings for each comparison and outcome
- (9) presenting the limitations of the synthesis.

See <https://www.bmj.com/content/368/bmj.l6890>

We will consider the number of studies reporting associations in the same direction between similar/comparable measures of exposure and outcomes and present the range of the magnitude of the reported associations, with assessment of any important study design factors that may account for differences in the magnitude of association (e.g. adjustment or population characteristics).

Where outcome measures within the same domains are not comparable, we will summarise the broad direction and magnitude of associations across the domain, reflecting on important drivers of the heterogeneity.

9. I am wondering how this study can answer the factors associated with the difference of health outcomes and healthcare use – based on the systematic review data.

Thank you for this question. As the primary form of summarising results will be narrative synthesis, we will follow the SWiM guidelines (see response 8 above) for synthesis without meta-analyses to summarise findings on moderating factors of the effect of single-parenthood on child health. We will first consider the rationales for studies as well as rationales for the analyses conducted in each study, reporting on effect modifiers that have been used to explore associations in the data. With this we can develop hypotheses of how parental or family characteristics may affect the association between single parenthood and child health.

If we determine that meta-analyses are possible, the effect of any reported effect modifiers will be tested using sub-group analysis (for categorical variables) or meta-regression (for continuous variables).

Reviewer: 2

Many thanks for the opportunity to review this systematic review protocol. Increasing understanding of the mechanisms linking lone parenthood to less favourable child outcomes is a very important research question, and I am glad to see that a review of this topic is being conducted. The protocol is well written and the research questions are clear. I have several comments below which I hope will help to improve the protocol.

1. The authors note that health is better in children with continuously married parents than for children with a single parent or separated parents. However, it is common to transition in and out of relationships, usually with new partners. Will outcomes for children in such households be considered, or will study participants be defined in terms of having only one biological parent in the household regardless of subsequent relationship transitions?

Thank you for this question. At this stage we have decided to exclude studies which solely focus on changes or transitions in family structure/parental marital status, for example, studies examining the child health effects of divorce. However, we appreciate this point and have included mention of parental relationship breakdown in the introduction (p4, lines 152-153 / 160-162). We have also clarified the scope of our systematic review in the inclusion criteria (p6, lines 273-278)

2. The authors note that poverty is higher among single parent families, but there is no discussion of

why this is the case.

We have now described in more detail the relationship between single parenthood and poverty in the 4th paragraph of the introduction (p3, lines 111-121). Poverty in single-parent families is linked to factors like wage gaps between men and women, job insecurity and lower earning potential for example.

3. Higher poverty and less time with children are mentioned as just two of a range of possible mechanisms linking lone parenthood to adverse child outcomes in the abstract. The background section focuses on the role of parental SES both prior to and after becoming single parents. However, the literature has identified many other potential mechanisms linking lone parenthood with adverse child outcomes, including lack of social support, stigma, role strain, and unemployment. Studies I am aware of tend to find that negative health outcomes are not entirely attenuated when SES is controlled for, leaving some uncertainty about the other factors contributing to poor outcomes. Some discussion of the range of potential mechanisms, and the nature of the evidence for them, would be helpful for framing the review.

Thank you for this comment, we agree. One of the initial motivations for the review was to better understand the extent to which factors other than SES explain the association between single parenthood and child outcomes. We have now mentioned other potential mechanisms linking single parenthood to adverse child outcomes including parental stress, lack of social networks and support and social stigma (p3, lines 124-129).

4. The authors note the challenges involved in studying lone fathers, but do not explicitly state whether studies of lone fathers will be included in the review. Given that they seem to be quite a distinct population, would it be appropriate to combine studies of lone mothers and lone fathers?

We agree that it may not be appropriate to combine these studies. However, we aim to include studies comparing any single parents (mother or father) to coupled parents, as described in the inclusion criteria (p5, line 256). In the synthesis section, we have now described that we will summarise how single parents are defined and that we will separate single mothers and single fathers in sub-group analyses (p7, lines 347-355).

5. Again, the authors refer to the difficulty of defining and sometimes identifying lone parents, and how different types of lone parent family may influence outcomes, but do not discuss how they will approach this in the studies. Will an attempt be made to establish how LPs are defined in each of the studies, and if so whether/how these differences do impact on child outcomes? For instance, as far as I am aware the USA has historically defined cohabiting mothers as single parents, whether they reside with the biological father or a new partner. In Sweden, joint custody is now the default position, and it is rather difficult to identify who is the 'lone parent' in such circumstances. Differences such as this would have to be considered when comparing outcomes across studies from different countries.

Yes, we aim to summarise findings with as much detail as possible, based on the information provided within the included studies. We have emphasised the fact that some definitions of single parents could include cohabiting couples in the second last paragraph of the introduction (p4, lines 168-173). We have also amended the text in the synthesis section of the protocol to make it clearer that we will categorise studies based on how single parents are defined (p7, lines 347-350).

6. Was an information scientist involved in developing the search strategy? I would suggest including the term 'solo parent' as that is commonly used in Australia. I'm unsure whether it might be helpful to include terms for specific outcome measures (e.g. Child Behaviour Checklist etc.).

Thank you for this question. An information scientist was not involved but we had multiple discussions with librarians about the development of an appropriate search strategy. In light of your suggestion to include another term, we have rerun the searches for 'solo parent' AND the outcome terms to identify any results we may have missed. There were no results that would have been coded as inclusions during screening so we have decided to keep the existing search terms unchanged.

Regarding terms for specific outcome measures, we agree in principle that it might be helpful to include some additional terms but need to consider the already high number of results to be screened. One of the limitations of this search is the necessary trade-off between range of scope and what is feasible for our team. This will be included as a limitation when we write up the results.

7. Given that the likelihood of becoming a lone parent is much higher for people with lower SES, how do the authors intend to deal with causality in the included studies? I would assume studies which control for parental SES prior to becoming parents would be required to account for selection effects, although I am not personally aware of many studies which do so. There is also evidence that people with mental health issues are more likely to become lone parents, again leading to selection effects.

At the moment, the scope of the review focuses on describing associational studies; also, many of the studies will be cross-sectional. Our ability to comment on causality will be limited by the included studies and we will not be making any conclusions about causality based on the results of this review. However, we appreciate the reviewer's point and in the data extraction section on study design, we have added that we will extract information about whether studies make any causal claims (p6, line 314).

8. Related to this, there is no discussion of study design in the inclusion criteria. Will cross-sectional studies be included, or only longitudinal studies which can shed light on causality? Would qualitative studies be considered, as they can be particularly helpful for identifying mechanisms? Even if any study design will be included, it would be helpful to have some discussion about the advantages and limitations of different methods.

Thank you for these questions. We have specified in the inclusion criteria that only quantitative studies designs will be included (p6, lines 286-290). Qualitative data will not be included in the analyses of this review but will be used to inform the discussion section. We will identify relevant qualitative work that might help explain conclusions from this systematic review.

9. There is no mention of geographical inclusion criteria. Will studies from any country be included or will they be restricted to high-income countries? Cultural and contextual differences with lower income countries may be quite challenging to account for in the synthesis.

At the moment we will include studies from any country, extracting information on study setting (as described on page 6, lines 311-313). If this information is available from included studies, we will do sub-group analyses as described in the synthesis section (p7, lines 354-358).

10. Do the authors intend to include outcomes such as substance abuse, self-harm or suicidality in the mental health domain?

Yes, we will include such outcomes in the mental health outcomes group and have now specified this in the outcomes section of the protocol (p6 line 302-303).

“Mental health and development outcomes: including disruptive behaviour, substance abuse, anxiety or depressive disorders, autism-spectrum disorders, psychosis, self-harm and suicidality, cognitive abilities (problem solving, memory, language/communication, early years educational attainment), social-emotional development (personal-social skills)

11. How will contextual differences be dealt with? Some studies have shown that outcomes for lone parents vary across countries with differing welfare regimes, and also that economic and labour market changes can influence outcomes for lone parents. These provide insights into possible mechanisms linking lone parenthood and other outcomes, and indeed potential intervention points for addressing adverse outcomes.

In line with your comment 9 above and comments 3/6 from reviewer 1, we agree with the reviewers' points and have now included mention of country context in the data extraction section (p6 lines 311-312) and the data synthesis section (p7, lines 354-358). Where available, this data will inform our synthesis of potentially protective factors. Any indication of contextual differences beyond the country and time period is unlikely to be reported within included studies and is outside the scope of this review.

12. I'm not clear why a risk of bias tool developed for systematic reviews is being used to inform the risk of bias assessment. Perhaps the ROBINS-I tool would be helpful for assessing non-randomised studies.

Thank you for pointing this out; we have amended this accordingly (p7, line 334).

13. Weighted mean differences will be used for continuous outcomes. Given that reported outcomes such as depression are likely to use heterogeneous scales, I would suggest that standardised mean differences would be more appropriate. These can also permit inclusion of several heterogeneous outcomes from single studies in forest plots even where meta-analysis is not possible.

Thank you for this suggestion; we have amended this accordingly (p8, line 399-401).

14. NB line 26 'four' should read 'five'.

Thank you for catching this typo (p7, line 332).